# Assessment of Knowledge, Attitude and Practice of Food Labeling and Expiry Date among the Female Health Sciences Students: A Public Health Concern

Fatima Riaz [1], Amna Moiz [2], Syed E. Mahmood [1,*], Ausaf Ahmad [3], Shahabe Saquib Abullais [4] and Shafait Ullah Khateeb [5]

1. Department of Family and Community Medicine, College of Medicine, King Khalid University, Abha 62529, Saudi Arabia; fatima.riaz786@yahoo.com
2. Medical City, King Khalid University, Abha 62529, Saudi Arabia; dr_amnamoiz@yahoo.com
3. Department of Community Medicine, Integral Institute of Medical Science and Research, Integral University, Lucknow 226026, India; ausaf.ahmad86@gmail.com
4. Department of Periodontics and Community Dental Sciences, College of Dentistry, King Khalid University, Abha 62529, Saudi Arabia; sshahabe@kku.edu.sa
5. Department of Restorative Dental Sciences, College of Dentistry, King Khalid University, Abha 62529, Saudi Arabia; skhateeb@kku.edu.sa
* Correspondence: smahmood@kku.edu.sa; Tel.: +966-550484344

**Abstract:** Nutrition information on food labels can help consumers to choose healthier food. We investigated consumers' awareness of food labels and their influence on the decision to buy food items among students of health sciences of King Khalid University, Abha. This cross-sectional study involved 350 females who gave informed verbal consent and were selected by systematic random sampling technique. Data was collected by using a self-administered questionnaire. Statistical analysis was done by using SPSS version 21. Overall 76.3% of students knew food labeling. Significant differences ($p < 0.05$) were observed between the knowledge about food labeling and education, the number of family members, earning members, frequency of shopping and income. Almost half of the students checked food labels before buying, 43.7% replaced food on an importance basis and value of labeling, and 60% replaced on a cost basis. More than half of students were ready to buy food items with no labels, and 21.7% even utilized expired food items because of a lack of knowledge regarding expiry date and low cost of food which could be hazardous for their health. Awareness of food labeling and expiry date should be enhanced by including this subject in the curriculum and electronic media to avoid health hazards of expired food items. Choosing healthy food options shall reduce the nutrition and chronic diseases among the general population in future.

**Keywords:** food labeling; expiry date; calories; nutritional value; females; Saudi Arabia

## 1. Introduction

Food and health safety are interrelated with each other. In today's modernized world use of packaged food items is tremendously increased. Therefore, the importance of food labeling has also increased. Nowadays food product labeling has become a popular policy tool in the food industry. Nutrition labeling is a valuable tool particularly when we talk about the provision of nutrition and health information and helps in learning how to apply nutrition information practically. The nutrition information on food labels can help consumers to choose healthier food [1].

An increasingly interconnected global food supply chain means that the risks posed by unsafe food items have also increased which imposes the potential to rapidly evolve from a local problem to an international emergency. Ensuring food safety is an essential component of achieving global health security and national food safety authorities must be able to share information quickly and efficiently worldwide [2].

The role of food labels is to inform consumers about the quality of food and help the seller in selling their product. The importance of food labeling has greatly increased and has become complex in the terms of food legislation, food companies, retailers, public authorities, and the consumer. According to the World Health Organization (WHO), the definition of food labeling a label includes any written, printed, or graphical material that should be present on the label of a food product, or is displayed near the food, including that to promote its sale or disposal [3].

In Saudi Arabia Saudi food and drug authority (SFAD) regulates and monitor foods, drugs, medicines, and diagnostic devices as well with the implementation of compulsory standards, testing them in laboratory and taking care of consumer awareness. It deals with food packaging, labeling and expiry dates as well. Their main objective is the safety of food for the consumption for humans and regulates its effectiveness [4].

Food labeling is one of the important processes in the packaging of food items which is usually overlooked by most consumers. The label is the first point of contact between the buyer and the manufacturer. Food labels contain a lot of information including calories, net quantity, expiry dates, shelf life, nutritional information, manufacturer's name, brand, price, instructions to use, and contents of food items used in the product which can help consumers to decide what to choose and what not to choose, etc. Food labels are also used to identify different food products and used to discriminate products from one to another so that consumers can decide to purchase them. In general, food labels give information to the consumers about the composition and the nature of food products to avoid confusion and protect the consumers against misuse of a particular food item, health risks, and its abuse. Marketing information, including the selling price, brand name, and commercial offers, is provided as well as information on the safe storage, preparation, and handling of the food product [5].

It is also evident from the various researches that food labeling is an important way by which consumers can get awareness regarding food items that they wanted to buy [3,5–7].

Hence knowledge, attitude and practices (KAP) surveys measure the level of awareness of consumers, their attitude towards buying food items, and measure their habits of practicing food labeling regarding purchasing food items that are directly affecting their health status. These studies also help in finding the gaps between knowledge and practices of certain behavior.

KAP studies tell us what people know about certain things, how they feel, and how they behave. Each study is designed for a specific purpose, research question, and setting. Food labeling information and consumers' level of education impose a large contribution in ensuring consumers about the good and bad effects of food items which they consumed [1]. Although many studies from the Middle East have explored the subject of consumers' awareness of food labels in different population groups however a study on food labels awareness among health science students in the Aseer region of the Kingdom of Saudi Arabia not much explored [8–11]. Whereas a study conducted in Riyadh among female students showed the 17.7% frequency of use of food by participants where as 16.8% of respondents never use food labeling before purchasing food items [11]. Therefore, we conducted this study in Aseer region to see the level of knowledge, attitude and practices of female health sciences students about food labeling and expiry dates. Our study hypothesis is that better awareness and attitude influence good practices regarding food labeling and expiry dates among the consumers. We assumed that health sciences students are aware of food labeling and also practices accordingly before purchasing food items.

The health sciences students being role models in society can play an important role in enhancing the awareness of food labels and expiry dates in the general population in the future. They are in closer contact with patients and their families regarding managing their health issues and giving health education. Therefore, providing health education to the general population ultimately results in improving the quality of health for the addressing population, hence decreasing the burden of many problems and issues caused by consuming expired food items and the hazards of not knowing the content of food items.

If their knowledge is better, they can reflect it in a good way and convey their message rightly to the patients and their families. Therefore, they can bring about a great reform in the community.

With this background, we undertook this study to investigate consumers' awareness of food labels and their influence on the decision to buy food items in the study population.

## 2. Materials and Methods

### 2.1. Study Area and Population

Abha is one of the beautiful cities of Saudi Arabia in the province of the Aseer region which is situated on the slopes of the Sarawat Mountains. This cross-sectional study was conducted among the female undergraduate health sciences students from the faculties of medicine, nursing, and dentistry of King Khalid University, Abha to assess their awareness of food labels and their influence on the decision to buy food items.

We conveniently conducted this study at the female section of the King Khalid University exclusively therefore we only recruit female students of health sciences colleges. Data was also collected by female staff only. Data collection was done from January 2014 to December 2017 by the principal investigator herself to prevent observer bias.

### 2.2. Inclusion and Exclusion Criteria

Any health science female student currently studying at a King Khalid University above 19 years was considered to be eligible to take part in the study. Those students who agreed to enroll in the study and gave informed consent regarding participation in the study were included. The student's confidentiality and anonymity were maintained, and all the information was gathered by the principal investigator herself.

### 2.3. Ethical Approvaland Data Collection

This study was approved by the Research Ethical Committee of the College of Medicine, King Khalid University. All the information was gathered by the principal investigator herself. The data was collected through a self-administered questionnaire distributed by direct contact with students. A pilot study was conducted with over 30 students, questionnaires obtained from the pilot study were analyzed but not included in the study and the study questionnaire was modified according to the findings of the pilot study.

Data was collected during the free period after finishing students' classes so that their studies would not be affected or disturbed because of the data collection procedure. Questionnaires were distributed to selected samples altogether in one class after taking permission from their teacher and fulfilling other requirements. Questionnaires were collected back after 30 min. Investigator made sure that all information was filled in by students properly. A 41-item questionnaire was adopted by the principal author after conducting an extensive review of the literature [12–14] and based on instruments used in previous studies. It comprised of four parts: The first part consisted of 8 questions related to the information on demographic, socioeconomic, health-related data; the second part consisted of 13 questions related to knowledge about food labeling and expiry date; the third part comprised of 8 questions regarding attitude and fourth part consisted of 12 questions investigated the responder's practice regarding food labeling and expiry date. Participants were given "Yes" and "No" option questions or multiple responses to closed-ended questions.

### 2.4. Sample Size and Sampling Technique

The study sample size was estimated using the Raosoft sample size calculator. A total study population size of 750 using a 50% response distribution for the largest sample size (n = 350) was obtained at a margin of error of 3.83%, a 95% confidence interval (CI) from the faculties of medicine, nursing, and dentistry of King Khalid University. By systematic random sampling techniques, out of the total 750 health science students every

2nd student was selected hence a total of 350 students selected and questionnaires were filled by the students.

### 2.5. Statistical Analysis

Data was entered, and statistical analysis was done by using a statistical package for social sciences (SPSS version 21). The frequency distribution table was used for categorical variables to present the knowledge of respondents about food labeling and food nutritional value. Under descriptive statistics, Pie and bar diagrams were used for the representation of categorical outcomes. Tests of significance like the Chi-square test are applied to find out the statistical significance of the difference in percentages. Univariate analysis was done using respondent knowledge about food labeling and reason for buying expired food as the dependent variable and the sociodemographic and behavioral factors were identified as independent variables. $p$-value of < 0.05 was taken as statistically significant for the calculations of variables.

### 3. Results

Out of the 350 students selected, a higher proportion (76.3%) was knowing food labeling and the remaining (23.7%) were not knowing.

In the age group 22–24 years, the percentage of knowledge was slightly lower as compared to the age group 19–21 years. The majority of students were in 19 to 21 years. The percent of knowledge among married students was higher compared to the number of single students. Hundred percent of subjects had knowledge about food labeling, in the group who had one kid. Medical students (83.9%) were having higher knowledge followed by dentistry (72.7%) and nursing students (70.2%). In the majority of families, the father was earning member, and 80.7% of fathers were having knowledge about food labeling. Most of the students prefer to go shopping every weekend, of which 66.5% were having knowledge about food labeling. More than 200 students were belonging to greater than 10,000 income levels. The differences observed among the education, number of family members, number of earning members, frequency of going shopping and income, and the knowledge about food labeling were found statistically significant ($p$ < 0.05) (Table 1).

**Table 1.** Distribution of respondents according to knowledge about food labeling.

| Characteristics | | | Knowledge about Food Labeling | | Total | Chi-Square $p$-Value |
|---|---|---|---|---|---|---|
| | | | Yes | No | | |
| Age groups | 19–21 years | N | 240 | 74 | 314 | 0.03, 0.84 |
| | | % | 76.4% | 23.6% | 100.0% | |
| | 22–24 years | N | 27 | 9 | 36 | |
| | | % | 75.0% | 25.0% | 100.0% | |
| Marital Status | Single | N | 246 | 79 | 325 | 0.88, 0.34 |
| | | % | 75.7% | 24.3% | 100.0% | |
| | Married | N | 21 | 4 | 25 | |
| | | % | 84.0% | 16.0% | 100.0% | |
| Number of Kids | 0 | N | 256 | 81 | 337 | 4.35, 0.11 |
| | | % | 76.0% | 24.0% | 100.0% | |
| | 1 | N | 9 | 0 | 9 | |
| | | % | 100.0% | 0.0% | 100.0% | |
| | 2 | N | 2 | 2 | 4 | |
| | | % | 50.0% | 50.0% | 100.0% | |

**Table 1.** *Cont.*

| Characteristics | | | Knowledge about Food Labeling | | Total | Chi-Square p-Value |
|---|---|---|---|---|---|---|
| | | | Yes | No | | |
| Education | Nursing students | N | 80 | 34 | 114 | |
| | | % | 70.2% | 29.8% | 100.0% | |
| | Medical students | N | 115 | 22 | 137 | 7.48, 0.02 |
| | | % | 83.9% | 16.1% | 100.0% | |
| | Dentistry students | N | 72 | 27 | 99 | |
| | | % | 72.7% | 27.3% | 100.0% | |
| Number of family members | Upto 2 | N | 6 | 15 | 21 | |
| | | % | 28.6% | 71.4% | 100.0% | |
| | 3–5 members | N | 33 | 20 | 53 | 38.28, 0.00 |
| | | % | 62.3% | 37.7% | 100.0% | |
| | 6 and more | N | 228 | 48 | 276 | |
| | | % | 82.6% | 17.4% | 100.0% | |
| Number of earning members | Father | N | 209 | 50 | 259 | |
| | | % | 80.7% | 19.3% | 100.0% | |
| | Mother | N | 13 | 0 | 13 | |
| | | % | 100.0% | 0.0% | 100.0% | 43.23, 0.00 |
| | Both Parents | N | 28 | 33 | 61 | |
| | | % | 45.9% | 54.1% | 100.0% | |
| | Siblings | N | 17 | 0 | 17 | |
| | | % | 100.0% | 0.0% | 100.0% | |
| Frequent of going shopping | Twice a week | N | 7 | 13 | 20 | |
| | | % | 35.0% | 65.0% | 100.0% | |
| | Every weekend | N | 109 | 55 | 164 | |
| | | % | 66.5% | 33.5% | 100.0% | 47.38, 0.00 |
| | Monthly | N | 86 | 9 | 95 | |
| | | % | 90.5% | 9.5% | 100.0% | |
| | Occasionally | N | 65 | 6 | 71 | |
| | | % | 91.5% | 8.5% | 100.0% | |
| Income (in SAR) | Up to 5000 | N | 71 | 0 | 71 | |
| | | % | 100.0% | 0.0% | 100.0% | |
| | 6000–10,000 | N | 60 | 17 | 77 | 31.14, 0.00 |
| | | % | 77.9% | 22.1% | 100.0% | |
| | >10,000 | N | 136 | 66 | 202 | |
| | | % | 67.3% | 32.7% | 100.0% | |

Nearly 76.3% of students were knowing the importance of food labeling. The source of information for the majority of students was lectures (33.4%) followed by lectures and internet (23.1%) and electronic media (22.6%). Only 2.3% of students got information from shopkeepers. Seventy-six percent of students were knowing the importance of the nutritional value of food (Table 2).

Out of 350 students, approximately half of the students checked all the food labels before buying. Nearly 43.7% of students replaced their items based on the importance and value of food labeling. About 60% of students replaced food items based on cost. More than half of the students were ready to buy food items with no food labels. Place of manufacturing of food items was not an important consideration for students before buying them. The majority (83.7%) of students were concerned to check the expiry dates of every food item. More than 90% of students had practiced checking expiry dates because they knew that expired food is not good for their health. About 78.3% of students were not buying anything expired if available at a low cost or for free. Most consumers used shopping lists (86.3%) and were attracted to buying food items at a low price (64%). Nearly 60.6% of students had a preference to buy branded food items. (Table 3).

**Table 2.** Knowledge of study subjects about food labeling and food nutritional value.

| Variables | Response | N | % |
|---|---|---|---|
| Knowledge about the importance of food labeling | Yes | 267 | 76.3 |
| | No | 83 | 23.7 |
| Source of information about food labeling | Lectures | 117 | 33.4 |
| | Print media | 27 | 7.7 |
| | Electronic Media | 79 | 22.6 |
| | Internet | 38 | 10.9 |
| | Shop Keeper | 8 | 2.3 |
| | Lectures and internet | 81 | 23.1 |
| Knowledge about the importance of food nutritional value | Yes | 266 | 76.0 |
| | No | 84 | 24.0 |
| Total | | 350 | 100.0 |

**Table 3.** Distribution of subjects based on practice in food labeling.

| Variables | Response | N | % |
|---|---|---|---|
| Food labeling | | | |
| Do you check all the food labels which you buy? | Yes | 167 | 47.7 |
| | No | 183 | 52.3 |
| Do you replace items on basis of the importance and value of food labeling? | Yes | 153 | 43.7 |
| | No | 197 | 56.3 |
| Do you replace food items on basis of cost? | Yes | 208 | 59.4 |
| | No | 142 | 40.6 |
| Do you buy food items with no food labels? | Yes | 193 | 55.1 |
| | No | 157 | 44.9 |
| What do you see on a food label? | Calories | 41 | 11.7 |
| | Expiry Date | 167 | 47.7 |
| | Place of Manufacturing | 9 | 2.6 |
| | Calories + Nutritional Value | 54 | 15.4 |
| | Calories + Nutritional Value + Expiry Date | 53 | 15.1 |
| | Nutritional Value + Expiry Date | 26 | 7.4 |
| Expiry date | | | |
| Do you check the expiry dates of every food you buy? | Yes | 293 | 83.7 |
| | No | 57 | 16.3 |
| Do you check expiry dates because you know that expired food is not good for your health? | Not Good For Health | 332 | 94.9 |
| | Don't Know | 18 | 5.1 |
| Do you buy anything expired if available at low cost or free? | Don't Buy | 274 | 78.3 |
| | Buy | 76 | 21.7 |
| Characteristics for buying food items | | | |
| Does the consumer use a shopping list? | Yes | 302 | 86.3 |
| | No | 48 | 13.7 |
| Do low prices attract you to buy food items? | Yes | 224 | 64.0 |
| | No | 126 | 36.0 |
| Do you prefer to buy branded food items? | Yes | 212 | 60.6 |
| | No | 138 | 39.4 |
| Total | | 350 | 100.0 |

Pie chart in Figure 1 shows the attitude of students towards nutritional information included on the food labels. Out of 350 students, 82% thought that the "nutrition information" should be included on the food package label.

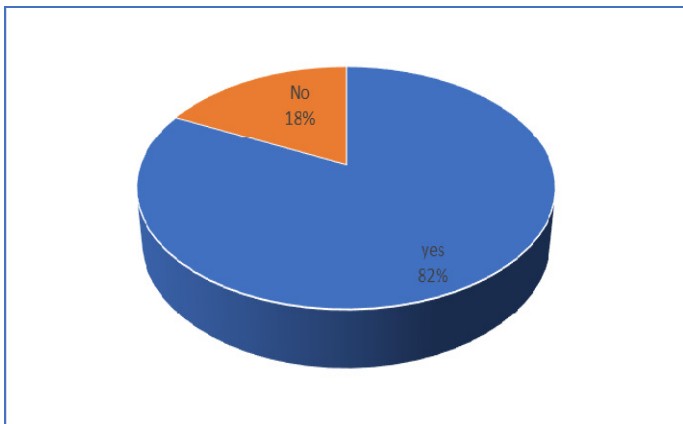

**Figure 1.** Distribution of subjects on the basis of attitude toward nutrition information included on labels.

Vertical graphs in Figure 2 depicts that the majority of students (78.3%) do not buy anything expired if available at a low cost or for free. However, some of the students (21.7%) were still utilizing these products, in the majority of the students (7.4%) were buying and using expired products because they don't know the hazards of expired food. About 4.9% of students were buying and using expired products because of their low cost. Very few students (2.9%) were not aware that food is expired.

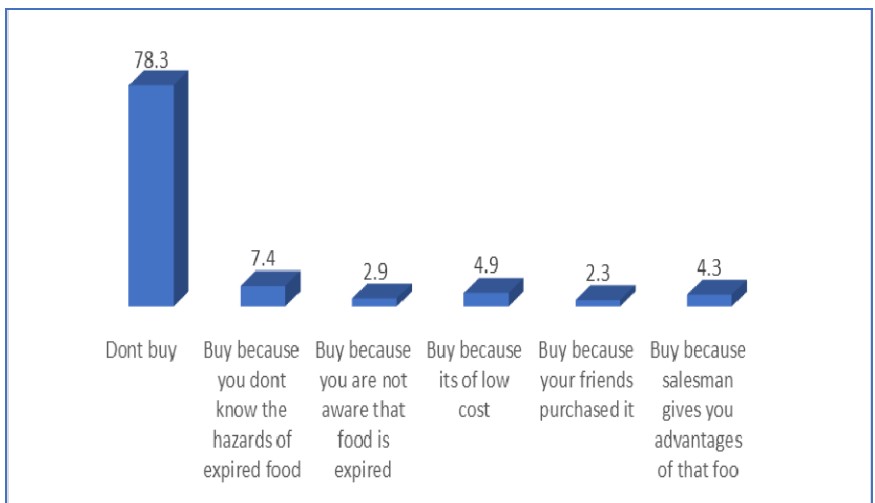

**Figure 2.** Distribution of subjects on the basis of practice about buy anything expired if available at low cost or free.

A higher percent of medical (89.1%) followed by dentistry (88.9%) and nursing students (56.1%) were not buying anything expired if available at a low cost or free. Attention regarding expiry products became more prominent with the increase in the number of family members. About 7.3% (19/259) of the students with earning fathers didn't know the hazards of expired food while 3.1% (8/259) were not aware that the food was expired. Nearly 61.5% (8/13) of students whose mothers were earning, buying anything expired if available at low cost or free as compared to 1.9% (5/259) of earning fathers. A higher proportion of those who had an income >10,000 SAR (78.3%) had more concern about not buying anything expired followed by those earning 6000–10,000 SAR (72.7%) and up to 5000 (66.2%) respectively (Table 4).

**Table 4.** Association with the buying of expired food with socio-demographic variables.

| Socio-Demographic Variables | | | Buying Anything Expired if Available at a Low Cost or Free | | | | | | Total | Chi-Square, *p*-Value |
|---|---|---|---|---|---|---|---|---|---|---|
| | | | Don't Buy | Reason for Buying | | | | | | |
| | | | | Don't Know the Hazards of Expired Food | Not Aware That Food Is Expired | It's of Low Cost | Friends Purchase It | Salesman Gives Advantages of Expired Food | | |
| Education | Nursing students | No | 64 | 14 | 3 | 15 | 4 | 14 | 114 | 68.15, 0.00 |
| | | % | 56.1% | 12.3% | 2.6% | 13.2% | 3.5% | 12.3% | 100.0% | |
| | Medical students | No | 122 | 5 | 5 | 1 | 3 | 1 | 137 | |
| | | % | 89.1% | 3.6% | 3.6% | 7% | 2.2% | 7% | 100.0% | |
| | Dentistry students | No | 88 | 7 | 2 | 1 | 1 | 0 | 99 | |
| | | % | 88.9% | 7.1% | 2.0% | 1.0% | 1.0% | 0% | 100.0% | |
| Number of family members | 2 & below | No | 8 | 6 | 0 | 3 | 1 | 3 | 21 | 32.90, 0.00 |
| | | % | 38.1% | 28.6% | .0% | 14.3% | 4.8% | 14.3% | 100.0% | |
| | 3–5 members | No | 47 | 2 | 2 | 2 | 0 | 0 | 53 | |
| | | % | 88.7% | 3.8% | 3.8% | 3.8% | 0% | 0% | 100.0% | |
| | 6 & more | No | 219 | 18 | 8 | 12 | 7 | 12 | 276 | |
| | | % | 79.3% | 6.5% | 2.9% | 4.3% | 2.5% | 4.3% | 100.0% | |
| Number of earning members | Father | No | 213 | 19 | 8 | 5 | 7 | 7 | 259 | 0.01, 0.00 |
| | | % | 82.2% | 7.3% | 3.1% | 1.9% | 2.7% | 2.7% | 100.0% | |
| | Mother | No | 5 | 0 | 0 | 8 | 0 | 0 | 13 | |
| | | % | 38.5% | 0% | 0% | 61.5% | 0% | 0% | 100.0% | |
| | Both Parents | No | 39 | 7 | 2 | 4 | 1 | 8 | 61 | |
| | | % | 63.9% | 11.5% | 3.3% | 6.6% | 1.6% | 13.1% | 100.0% | |
| | Siblings | No | 17 | 0 | 0 | 0 | 0 | 0 | 17 | |
| | | % | 100.0% | 0% | 0% | 0% | 0% | 0% | 100.0% | |
| Income (in SAR) | up to 5000 | No | 47 | 2 | 2 | 9 | 3 | 8 | 71 | 41.39, 0.00 |
| | | % | 66.2% | 2.8% | 2.8% | 12.7% | 4.2% | 11.3% | 100.0% | |
| | 6000–10,000 | No | 56 | 13 | 1 | 1 | 2 | 4 | 77 | |
| | | % | 72.7% | 16.9% | 1.3% | 1.3% | 2.6% | 5.2% | 100.0% | |
| | >10,000 | No | 171 | 11 | 7 | 7 | 3 | 3 | 202 | |
| | | % | 84.7% | 5.4% | 3.5% | 3.5% | 1.5% | 1.5% | 100.0% | |
| Total | | No | 274 | 26 | 10 | 17 | 8 | 15 | 350 | |
| | | % | 78.3% | 7.4% | 2.9% | 4.9% | 2.3% | 4.3% | 100.0% | |

## 4. Discussion

The food nutrition label provides the nutrition information that helps consumers with food choices and is used to give information so that customers can choose between foods. The present study was performed to assess awareness of food labels and their influence on the decision to buy food items among female consumers. This study shall provide information on the awareness of labeled food information among health sciences students of King Khalid University Abha, KSA. The information will bring attention to the policymakers on the need of designing and conducting programs to improve consumers' awareness regarding food labeling information and its influence on purchasing food items from the markets. The evidence generated from this study could help experts in the field of nutritionists, science laboratories, the health personnel, and standard organization of the country which will go a long way in preventing some of the preventable diseases that have claimed lives of thousands due to lack of understanding of the labeling on the goods purchased.

### 4.1. Knowledge of Food Labeling and Expiry Date

The differences observed among the education, number of family members, number of earning members, frequency of going shopping and income, and knowledge about food labeling were found statistically significant ($p < 0.05$).

Our findings are comparable to the Nigeria study which revealed a correlation between the level of knowledge and attitude towards food labels on one hand and the level of education, geographical location, and socio-economic standing of consumers on the other [15]. On the contrary, demographic characteristics of respondents had non-significant relation in a study conducted elsewhere [16].

In this study, the majority (76.3%) of students were knowing the importance of food labeling which could be because the respondents were all undergraduate students and had had prior knowledge of food labels.

In the literature it is also found that highly educated people are more expected to use food labels [17]. On the other hand, Hiew et al. reported that the study subjects with at least a diploma had a broadly high level of knowledge of nutrition information than those with a primary level education and better food choices [18].

The Source of information for the majority of students was lectures (33.4%) followed by lectures and internet (23.1%) and electronic media (22.6%). Only 2.3% of students got information from shopkeepers. The internet is a common source for obtaining information on food safety [19]. Students of health sciences are more knowledgeable about food labeling due to their curriculum which covers related topics also they are connected via technology, smartphone applications, and social media; therefore, they can easily access the web to obtain specific knowledge about food labeling.

The respondent's knowledge (76%) of the reported nutritional value of food in this study was relatively high which may be a result of health consciousness among the respondents and also probably influenced by their academic environment with free access to nutritional information. In a recent study conducted among Saudi adults, the knowledge regarding the nutrition facts label was about 52% and 62% of the Saudis declared that they always or sometimes used the nutrition facts label when purchasing food items [10].

In another cross-sectional study of Malaysian students, 21.6% of the 329 undergraduate students "often" use the food label during food purchasing decisions [20]. Our findings were divergent from the findings of a study conducted among college students by Norazmir et al. (2012) where a lack of knowledge regarding nutrition labels resulted in minimal food label reading [21].

Furthermore, in a study by Rachel Cooke, showed that nutritional label use mediates a good relationship between the knowledge of food labeling and their uses among consumers [22]. In another meta-analysis its being shown that most adults were ready to use information if it was provided on the label while the study subjects' views about the well-being of foods did not essentially depend on information on the label [23].

There is a discrepancy among studies reporting nutritional knowledge and its association with food which could be due to differences in sample size, ethnicity of the examined population, research methods used, study time, and other participant characteristics [18].

### 4.2. Attitude and Practice of Food Labeling and Expiry Date

Out of 350 students, approximately half of the students checked all the food labels before buying. Nearly 43.7% of students replaced their items based on the importance and value of food labeling. About 60% of students replaced food items based on cost. More than half of the students were ready to buy food items with no food labels. Place of manufacturing of food items was not an important consideration for students before buying them. The majority (83.7%) of students were concerned to check the expiry dates of every food item. More than 90% of students had a practice of checking expiry dates because they knew that expired food is not good for their health. About 78.3% of students were not buying anything expired if available at a low cost or for free. On the contrary, consumers were motivated to read labeling information when the product had a low price [16]. Most

consumers used a shopping list (86.3%) and were attracted to buying food items at a low price (64%). Nearly 60.6% of students had a preference to buy branded food items in this study.

In addition, Africa digital health repository project reported that about 32.5% stated that they look out for the total energy factor of the nutritional information on food labels. More than half (55.4%) reported that food labels affect their choices to purchase food products sometimes although 30.9% stated that food labels affect their choices to purchase food products always [24].

The findings of another study from India indicate that the majority (71.9%) of the participants claimed that they do not use a shopping list, and more than half of them (61.8%) indicated that their choice of specific foods was not based on nutrition information [5].

Out of 350 students, 82% thought that the "nutrition information" should be included on the food package label.

The majority of students (78.3%) do not buy anything expired if available at a low cost or for free. However, some of the students (21.7%) were still utilizing these products, in the majority of the students (7.4%) were buying and using expired products because they don't know the hazards of expired food. About 4.9% of students were buying and using expired products because of their low cost. Very few students (2.9%) were not aware that food is expired.

Psychological, economic, and social factors have been reported to play an essential role in influencing the use of nutrition labels in other studies [25].

Food labels are used to inform the consumers and help sell the food products. The food label is essential to help informed choices or to avoid false, confusing, or unreliable conduct [26]. Multiple factors are being measured during buying food products [27]. Some consumers search for extra information on a food package to aid them to relate the claim to their previous knowledge and practices [28], although some consumers consider food labels as a time-consuming thing and on the other hand, some consumers do not understand food label information well enough to make healthy food choices [29].

Several studies have focused on use of food labels in developed countries [30,31]. However, such studies are either scanty or non-existent in developing countries. Our study has a few limitations. The cross-sectional design cannot confirm the causality of the relationship between compared variables. The self-reported response could over or underestimate the result. The study's weakness is that it was conducted in a single city of the Aseer Region of KSA and involved only the female gender. We hope in the future to have all the required resources to do multicentric/nationwide studies. However, a representative sample including female students of medicine, dentistry, and nursing colleges is the strength of our study. Although, this study spans a long period of time, no academic intervention has ever been attempted to prevent the health hazards from the usage of expired food items among the study population, so it is less likely that our study observations are affected.

## 5. Conclusions

A vast majority of students were knowing food labeling and expiry date and they are not buying expired food items. However, more than half of the students were ready to buy food items with no food label and some of them were even utilizing expired food items because of a lack of knowledge regarding expiry date and low cost of expired food items which could be hazardous for their health. Hence this increases the need for public health awareness programs and reforms in curriculum regarding knowledge and the importance of food labeling and expiry date to avoid bad health outcomes. Therefore, it is strongly recommended that awareness of food labeling and expiry date should be enhanced by including this subject in the curriculum, public campaigns and seminars. Its importance should also be projected through mass media communication via electronic media, the internet, and social media. This will help to reduce the preventable health hazards from the usage of expired food items among the population.

**Author Contributions:** Conceptualization, F.R. and A.M.; methodology, F.R. and A.M.; software, A.A.; validation, F.R., A.A. and S.E.M.; formal analysis, A.A.; investigation, F.R. and A.M.; resources, S.S.A. and S.U.K.; data curation, S.E.M.; writing—F.R. and S.E.M.; writing—review and editing, S.E.M.; visualization F.R. and A.M.; supervision, S.S.A. and S.U.K.; project administration, S.S.A. and S.U.K.; funding acquisition, S.S.A.; All authors have read and agreed to the published version of the manuscript.

**Funding:** This research was funded by Deanship of Scientific Research at King Khalid University, Abha, Saudi Arabia, grant number RGP-2/234/1443 and The APC was funded by Deanship of Scientific Research at King Khalid University, Abha, Saudi Arabia.

**Institutional Review Board Statement:** The study was conducted according to the guidelines of the Declaration of Helsinki and approved by the Institutional Review Board (or Ethics Committee) of King Khalid University (REC# 2021-03-08).

**Informed Consent Statement:** Informed consent was obtained from all subjects involved in the study.

**Data Availability Statement:** Data is available on request.

**Acknowledgments:** The authors extend their appreciation to the Deanship of Scientific Research at King Khalid University, Abha, Saudi Arabia for funding this work through the Large Research Group Project under grant number (RGP-2/234/43).

**Conflicts of Interest:** The authors declare no conflict of interest.

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
