# Peer review of "Assessment of Knowledge, Attitude and Practice of Food Labeling and Expiry Date among the Female Health Sciences Students: A Public Health Concern"

_sustainability, doi:10.3390/su14116708_

Round 1
Reviewer 1 Report
Dear Authors,
The presented manuscript aimed to investigate consumers' awareness of food labels and their influence on the decision to buy food items in the study population.
The discussed subject is interesting, however, a big drawback is that this study was carried out almost 5 years ago. This begs the question, are those conclusions still up to date? You didn't specify what products did the analysis concern. It is necessary to add current research in this topic carried out by the Authors. Additionally, the cited literature regarding the study is outdated.
Specific comment on the manuscript is as follows:
- Why did only women participate in this study? It's not a representative group.
- It should be mentioned in the title that only women did participate in this study.
- The period between carrying out the study and submitting it to be published is too long.
- There are more up to date research findings. Example here:
Karen Hock, Rachel B. Acton, Alejandra Jáuregui, Lana Vanderlee, Christine M. White, David Hammond, Experimental study of front-of-package nutrition labels’ efficacy on perceived healthfulness of sugar-sweetened beverages among youth in six countries, Preventive Medicine Reports,Volume 24, 2021,101577,https://doi.org/10.1016/j.pmedr.2021.101577.
- Reference no 4 - lack of access date.
From my standpoint, this manuscript isn’t appropriate for publication in Journal - Sustainability, given the above aspects.
Author Response
Point 1: A big drawback is that this study was carried out almost 5 years ago. This begs the question, are those conclusions still up to date?
Explanation: As pointed out by the respected reviewer, this study was conducted 5 years ago but no academic intervention has ever been done in the university since then so it is less likely that our conclusion is affected because this time lag.
Point 2: You didn't specify what products did the analysis concern.
Explanation: Our analysis was concerned with all the type of food items available in packages containing food labels
Point 3: It is necessary to add current research in this topic carried out by the Authors. Additionally, the cited literature regarding the study is outdated.
Explanation: Current research articles has been added as suggested we have added the recent references.
Specific comment on the manuscript is as follows:
Point 4: Why did only women participate in this study? It's not a representative group.
Explanation: We conducted this study at the female section of the King Khalid University as per the objective of the research proposal and convenience in the sample collection. In our study and data was also collected by the concerned female staff only.
Point 5: It should be mentioned in the title that only women did participate in this study.
Explanation: As suggested we have modified the title.
“Assessment of knowledge, attitude and practice of food labeling and expiry date among the female health sciences students: A public health concern”
Point 6: It The period between carrying out the study and submitting it to be published is too long.
Explanation: We agree with the reviewer comments; however multiple factors are responsible for the delay as follows;
- Data collection period was prolonged because of systematic sampling technique.
- During COVID era the authors faced a lot of technical difficulties
- Because of lack of financial support, it could not get published earlier.
Point 7: It There are more up to date research findings. Example here:
Karen Hock, Rachel B. Acton, Alejandra Jáuregui, Lana Vanderlee, Christine M. White, David Hammond, Experimental study of front-of-package nutrition labels’ efficacy on perceived healthfulness of sugar-sweetened beverages among youth in six countries, Preventive Medicine Reports,Volume 24, 2021,101577,https://doi.org/10.1016/j.pmedr.2021.101577.
Explanation: The references has been updated as per the recommendation by the reviewer’s comments.
Point 8: Reference no 4 - lack of access date.
Explanation: Reference 4 has been updated with access date
Reviewer 2 Report
Dear authors,
I was pleased to read your manuscript "Assessment of knowledge, attitude and practice of food labeling and expiry date among the health sciences students: A public health concern”. The topic of your paper is one of a high importance, considering the role of food label as tool for orienting consumers’ food choices.
Below you will find my suggestions for improving the manuscript:
Abstract
The Abstract is too long, as it includes too much details about the results. Please try to shorten it, keeping references only about the main results.
Also, please add some information related to the possible practical implications of your research.
Introduction
In the first part of the Introduction, where you show the importance of the general research are, you should add some information (I suggest after line 82) about the mandatory nature of food label in some regions. For example, in the European Union, the content of food label (including the nutrition information) must be in compliance with EU Regulation 1169/2011. In the USA, nutritional label must comply to FDA requirements. Please indicate what is the situation in the Middle East and in the Kingdom of Saudi Arabia. Are there any government regulations that apply to food labeling? Is the declaration of nutrition information mandatory on food labels in the Kingdom of Saudi Arabia?
Line 83 – “It is also evident from the various researches that food labeling is an important way by which consumers can get awareness regarding food items that they wanted to buy [5]”. The statement is true, but you said “various research” and cited only one; please support that statement with other recent pieces of research in the area worldwide.
Line 85 - Please replace “KAP survey” with “Knowledge, Attitude and Practice (KAP) survey”, as this type of survey was firstly mentioned here.
After indicating the gap in previous research, in the Introduction section you should present the hypotheses of your research and indicate the methods used for data collection and data analysis.
Also, you should announce the principal findings/outcomes of your research.
The value of your research should be highlighted as well. The relevance of your study needs to be carefully indicated, either in relation to the gap identified or by statements of the contribution you intended to make.
In the end of the Introduction section, please consider to outline the structure of the paper (tell the reader what issues will be dealt with in every section of the article).
Literature Review
Because you have conducted a quantitative research, you should include a new section of Literature Review where you should clearly define the research hypotheses and support them by citing relevant sources in the field. The hypotheses must result from a critical analysis of the existing scientific literature.
The Literature Review section is meant to serve as an argument for the objectives and hypotheses that guide your research.
Materials and Methods
Please include at the beginning of the Materials and Methods section a restatement of your research purpose. Also, you should explain the appropriacy of the method (why did you choose this method?).
Line 115-116 – ”Any health science female student currently studying at a King Khalid University above 19 years was considered to be eligible to take part in the study”. Please explain why only female students were included in the sample.
Line 133-135 – ”A 41 item Questionnaire was adopted by the principal author after conducting an extensive review of the literature and based on instruments used in previous studies”. Please indicate few studies based on which you built the instrument for data collection.
Discussion
In this section you should explain how the results enable to accept or not each hypothesis of your research.
Conclusions
The Conclusions section should be rewritten. Please structure the section of Conclusions so as to show that the main purpose of your research was achieved. To do this, recall the main issues raised in the Introduction (especially the purpose of your study) and draw together the points made in the main body of the paper by emphasizing the implications of your findings. You should stress out what are the managerial implications of your research from social, economic and environmental point of view.
Author Response
Point 1: Abstract
The Abstract is too long, as it includes too much details about the results. Please try to shorten it, keeping references only about the main results. Also, please add some information related to the possible practical implications of your research.
Explanation: As suggested by the respected reviewer, abstract has been revised to shorten. Also practical implication has been added.
Point 2: Introduction
In the first part of the Introduction, where you show the importance of the general research are, you should add some information (I suggest after line 82) about the mandatory nature of food label in some regions. For example, in the European Union, the content of food label (including the nutrition information) must be in compliance with EU Regulation 1169/2011. In the USA, nutritional label must comply to FDA requirements. Please indicate what is the situation in the Middle East and in the Kingdom of Saudi Arabia. Are there any government regulations that apply to food labeling? Is the declaration of nutrition information mandatory on food labels in the Kingdom of Saudi Arabia?
Explanation: As suggested we have included the relevant information about The Saudi Food and Drug Authority. See page 2 para 4 Ref 4: https://old.sfda.gov.sa/en/food/about/Pages/overview.aspx.
Point 3: Line 83 – “It is also evident from the various researches that food labeling is an important way by which consumers can get awareness regarding food items that they wanted to buy [5]”. The statement is true, but you said “various research” and cited only one; please support that statement with other recent pieces of research in the area worldwide.
Explanation: As suggested, the following references have been cited in the tect
- Nadia P, Daleen van der Merwe*, Magdalena Bosman &Alet Erasmus A CRITICAL REVIEW OF THE SIGNIFICANCE OF FOOD LABELLING DURING CONSUMER DECISION MAKING. Journal of Family Ecology and Consumer Sciences, Vol 40, 2012
- Siphelele Vincent Wekeza and MelusiSibanda. Factors Influencing Consumer Purchase Intentions of Organically Grown Products in Shelly Centre, Port Shepstone, South Africa. Int. J. Environ. Res. Public Health 2019, 16, 956; doi:10.3390/ijerph16060956
- Koen N, MNutrition; Blaauw R, Wentzel-ViljoenE.Food and nutrition labelling: the past, present and the way forward. S Afr J ClinNutr 2016;29(1):13-21
Point 4: Line 85 - Please replace “KAP survey” with “Knowledge, Attitude and Practice (KAP) survey”, as this type of survey was firstly mentioned here.
Explanation: Correction done as suggested by the respected reviewer.
Point 5: After indicating the gap in previous research, in the Introduction section you should present the hypotheses of your research and indicate the methods used for data collection and data analysis.
Explanation: We have now presented the hypothesis that better awareness and attitude influence good practices regarding food labelling and expiry dates among consumers.
Point 6: Also, you should announce the principal findings/outcomes of your research.
Explanation: We assumed that health sciences students are aware of food labelling and also practices accordingly before purchasing food items.
Point 7: The value of your research should be highlighted as well. The relevance of your study needs to be carefully indicated, either in relation to the gap identified or by statements of the contribution you intended to make. In the end of the Introduction section, please consider to outline the structure of the paper (tell the reader what issues will be dealt with in every section of the article).
Explanation: The hypothesis of the research has been added and highlighted in the introduction section.
Literature Review
Point 8: Because you have conducted a quantitative research, you should include a new section of Literature Review where you should clearly define the research hypotheses and support them by citing relevant sources in the field. The hypotheses must result from a critical analysis of the existing scientific literature. The Literature Review section is meant to serve as an argument for the objectives and hypotheses that guide your research.
Explanation: The research hypothesis of the research has been added and highlighted in the introduction section. Relevant articles have been reviewed and information is updated to support the hypothesis.
Materials and Methods
Point 9: Please include at the beginning of the Materials and Methods section a restatement of your research purpose. Also, you should explain the appropriacy of the method (why did you choose this method?).
Explanation: As suggested this has been included. To minimize observer bias data collection was done by the principal investigator herself.
Point 10: Line 115-116 –” Any health science female student currently studying at a King Khalid University above 19 years was considered to be eligible to take part in the study”. Please explain why only female students were included in the sample.
Explanation: We conducted this study at the female section of the King Khalid University as per the objective of the research proposal and convenience in the sample collection. In our study and data was also collected by the concerned female staff only.
Point 11: Line 133-135 – ”A 41 item Questionnaire was adopted by the principal author after conducting an extensive review of the literature and based on instruments used in previous studies”. Please indicate few studies based on which you built the instrument for data collection.
Explanation: References 15, 16 and 17 have been added. The questionnaire was based on them.
Discussion
Point 12: In this section you should explain how the results enable to accept or not each hypothesis of your research.
Explanation: As per the recommendation of reviewer, the results are discussed under subheadings in the discussion section to accept of reject the hypothesis constructed for the research.
Conclusions
Point 13: The Conclusions section should be rewritten. Please structure the section of Conclusions so as to show that the main purpose of your research was achieved. To do this, recall the main issues raised in the Introduction (especially the purpose of your study) and draw together the points made in the main body of the paper by emphasizing the implications of your findings. You should stress out what are the managerial implications of your research from social, economic and environmental point of view.
Explanation: This has been modified as suggested
Reviewer 3 Report
This is a very interesting study. But the article is more like a research report and there are still some areas that need to be improved. For example, can the authors make a empirical study to assess the students’ awareness of food labels and their influence on the decision to buy food items?
- In the Introduction, the author used many separate paragraphs.But the basis and purpose of each paragraph cannot be seen. Especially why do the authors only investigate female students? It is suggested that the author combine some paragraphs in order to study the derivation of research gaps more hierarchically.
- In line 85, the abbreviation of KAP should be given the full name. In line 150, the heading doesnot need punctuation marks. The female consumer in the article should be female student consumer such as the expression in line 222.
- Did the author investigate 750 females or selected 350 females from 750 students? Why and how the indicators are selected? For example, the author analyzed the socio-demographic variables without introduce the source of its sub-indicators. The details of Materials and Methodsare insufficient to allow the work to be reproduced by an independent researcher.
- Thereis no subsection of Discussion, and the paper seems like a draft. It is suggested to add subheadings to reflect the results of this study. For example, the author can discuss the results from different aspects like knowledge, attitude and practice in food labeling.
Author Response
Point 1: In the Introduction, the author used many separate paragraphs. But the basis and purpose of each paragraph cannot be seen. Especially why do the authors only investigate female students? It is suggested that the author combine some paragraphs in order to study the derivation of research gaps more hierarchically.
Explanation: As suggested by respected reviewer, this is mentioned in the manuscript. King Khalid University has entirely separate campuses for boys and girls. We conveniently conducted this study at female section of the King Khalid University, so we did recruit only female students in our study and data also collected by female staff only.
Point 2: In line 85, the abbreviation of KAP should be given the full name. In line 150, the heading does not need punctuation marks. The female consumer in the article should be female student consumer such as the expression in line 222.
Explanation: Correction implemented as per the reviewer’s instruction.
Point 3: Did the author investigate 750 females or selected 350 females from 750 students? Why and how the indicators are selected? For example, the author analyzed the socio-demographic variables without introduce the source of its sub-indicators. The details of Materials and Methods are insufficient to allow the work to be reproduced by an independent researcher.
Explanation: As mentioned in the manuscript:
The study sample size was estimated using the Raosoft sample size calculator. A total study population size of 750 using a 50% response distribution for the largest sample size (n= 350) was obtained at a margin of error of 3.83%, a 95% confidence interval (CI) from the faculties of medicine, nursing, and dentistry of King Khalid University. By systematic random sampling techniques, out of the total 750 health science students every 2nd student was selected hence a total of 350 students selected and questionnaires were filled by the students.
Point 4: There is no subsection of Discussion, and the paper seems like a draft. It is suggested to add subheadings to reflect the results of this study. For example, the author can discuss the results from different aspects like knowledge, attitude and practice in food labeling.
Explanation: As suggested subheadings have been added. Also a few new references have been added.
Round 2
Reviewer 1 Report
Dear Authors,
Majority of previous comments have been taken into account.
I have only one minor comment to the revised text of the manuscript:
The reference No. 1, lines: 361-362, have a different format than the rest. Please, change it.
I haven’t more objections to the new version of the manuscript.
From my point of view the new version of the article is appropriate for publication in Journal - Sustainability, based on the above comment.
Author Response
As per suggestion of the Respected Reviewer we have corrected the format of the reference No. 1.
Thank you
Reviewer 2 Report
Authors made good efforts to adress all my concerns and I thank them for this.
Nevertheless, the conclusion section can still be improved.
Author Response
As per suggestions of the respected reviewer the conclusion section has been improved. The following paragraph has been added "Hence this increases the need for public health awareness programs and reforms in curriculum regarding knowledge and the importance of food labeling and expiry date to avoid bad health outcomes. Therefore it is strongly recommended that awareness of food labeling and expiry date should be enhanced by including this subject in the curriculum, public campaigns and seminars. Its importance should also be projected through mass media communication via electronic media, the internet, and social media. This will help to reduce the preventable health hazards from the usage of expired food items among the population."
Thank you
Reviewer 3 Report
Thank the authors for the efforts to improve the quality of the paper. However, the article still has some issues that need to be improved.
- The writer needs to check the entire text carefully to eliminate formatting problems.For example, in line 118 there is a period at the beginning of the sentence. And in line 157 the title is punctuated.
- The keyword setting is a bit puzzling and can be deliberated a bit more.
- The research spans a long period of time and what impact it may have. It is recommended that the authors indicate this appropriately in the outlook section.
Author Response
- As suggested by the respected reviewer we have made the required changes.
- The keyword setting has been revised.
- The following comment "This research spans a long period of time and what impact it may have. It is recommended that the authors indicate this appropriately in the outlook section." has been addressed. The following paragraph has been included " Although this study spans a long period of time, no academic intervention has ever been attempted to prevent the health hazards from the usage of expired food items among the study population so it is less likely that our study observations are affected."